# Activity of Bacteriophage D29 Loaded on Nanoliposomes against Macrophages Infected with *Mycobacterium tuberculosis*

**DOI:** 10.3390/diseases11040150

**Published:** 2023-10-26

**Authors:** Ana P. B. Silva, Cesar Augusto Roque-Borda, Christian S. Carnero Canales, Laura Maria Duran Gleriani Primo, Isabel C. Silva, Camila M. Ribeiro, Marlus Chorilli, Patrícia Bento da Silva, Joás L. Silva, Fernando Rogério Pavan

**Affiliations:** 1Tuberculosis Research Laboratory, School of Pharmaceutical Sciences, São Paulo State University (UNESP), Araraquara 14800-903, Brazil; 2Facultad de Ciencias Farmaceuticas, Bioquímicas y Biotecnológicas, Universidad Católica de Santa María, Arequipa 04000, Peru; 3Department of Genetics and Morphology of the Institute of Biological Sciences, University of Brasilia (UNB), Brasília 70910-900, Brazil; 4National Heart, Lung, and Blood Institute, National Institute of Health (NIH), Bethesda, MD 20892, USA

**Keywords:** intracellular activity, liposome, mycobacteriophage D29, *Mycobacterium tuberculosis*

## Abstract

The search for new antimicrobial agents is a continuous struggle, mainly because more and more cases of resistant strains are being reported. *Mycobacterium tuberculosis* (MTB) is the main microorganism responsible for millions of deaths worldwide. The development of new antimicrobial agents is generally aimed at finding strong interactions with one or more bacterial receptors. It has been proven that bacteriophages have the ability to adhere to specific and selective regions. However, their transport and administration must be carefully evaluated as an excess could prevent a positive response and the bacteriophages may be eliminated during their journey. With this in mind, the mycobacteriophage D29 was encapsulated in nanoliposomes, which made it possible to determine its antimicrobial activity during transport and its stability in the treatment of active and latent *Mycobacterium tuberculosis*. The antimicrobial activity, the cytotoxicity in macrophages and fibroblasts, as well as their infection and time–kill were evaluated. Phage nanoencapsulation showed efficient cell internalization to induce MTB clearance with values greater than 90%. Therefore, it was shown that nanotechnology is capable of assisting in the activity of degradation-sensitive compounds to achieve better therapy and evade the immune response against phages during treatment.

## 1. Introduction

The latest report from the World Health Organization has shown that *Mycobacterium tuberculosis* (MTB) is increasingly one step ahead of science since the incidence and infection rates have accelerated remarkably after the pandemic caused by SARS-CoV-2 [1]. High demand for new drugs has sidelined some excellent selective candidates for killing bacteria.

Phage therapy and technology against resistant bacteria has gained strength in recent years due to its specific activity against pathogenic bacteria that claim millions of lives each year [2,3]. To maintain their activity, phages seek to infect specific cells and inject genetic material, which allows them to be classified as lytic, lysogenic and pseudo-lysogenic [4]. The WHO launched a list of priority bacteria that cause many diseases and are of extreme urgency, including MTB [5]. Despite the increasing amount of research aimed at developing faster diagnostic tests and more effective drug treatments, MTB, the main pathogenic cause of tuberculosis (TB), remains one of the bacterial species that most afflicts humankind [6]. The increase in cases of patients with MTB resistant to rifampicin (RIF) is alarming, and many of the defense mechanisms of MTB lead to a longer treatment that is not very effective [7]. MTB in its latent form is characterized by hypoxia, low pH and the presence of nitric oxide and carbon monoxide, among other factors [8]. These latent MTB strains express various persistence genes, which makes them even more difficult to treat compared with active MTB [9].

Previous studies on mycobacteriophage D29 have demonstrated effective activity against MTB, but its administration is still a challenge. An important limitation for phage therapy is the survivability of MTB in macrophages [10] because free phages are inactivated by the immune system before the necessary direct contact with mycobacteria can be achieved, preventing lysis of mycobacterial cells [11,12]. Carrigy et al. [13] showed that the application of aerosol using a vibrating mesh nebulizer loaded with phage D29 was able to reduce the proliferation of MTB in the lungs (prophylactic protection). This treatment resulted in a significantly low bacterial load after one and three weeks after exposure. However, the method of phage delivery must be carefully chosen as it is important to generate a favorable treatment response. Carrigy et al. [14] had previously revealed that the jet nebulizer was not entirely efficient compared with the vibrating mesh nebulizer in animals. In fact, we still do not know how to insert bacteriophages into macrophages without their being easily inactivated.

Nanotechnology is a tool that allows the targeted delivery of drugs for a longer period [15,16]. It improves stability and manages to cross insurmountable barriers such as the intestinal barrier [17], macrophages [18] or the blood–brain barrier [19], allowing a better biodistribution of promising macromolecules [20]. Nanoliposomes are nanostructured transport systems for small molecules such as nucleic acids, phages and peptides [21]. They have a lipid structure that allows them to stealthily enter biological systems with wide global applications, including several approved by the USA and the EU [21,22]. Nieth et al. [23] studied giant liposomes loaded with mycobacteriophage TM4 and observed that liposome-associated phages are taken up into eukaryotic cells more efficiently than free phages, demonstrating the feasibility of using carriers for target delivery.

With the aim of achieving better administration of specific phages for Mycobacterium tuberculosis and avoiding the immune response, in this research article the use of a lipid-based carrier has been chosen, and we demonstrate that mycobacteriophage D29 on liposome nanoencapsulated systems has cellular internalization efficiency in infected macrophages and constitutes important vehicular administration against active and dormant MTB in a single dose.

## 2. Materials and Methods

### 2.1. Chemical Reagents

Soy phosphatidylcholine was purchased from Lipoid GmbH (Ludwigshafen, Germany). Magnesium sulfate (MgSO_4_), sodium chloride (NaCl), Tris-HCl, ether and oleic acid were purchased from Labsynth^®^ (Sao Paulo, Brazil). Middlebrook 7H9 Broth was obtained from Kasvi (Paraná, Brazil). Catalase was purchased from Thermo Fisher Scientific Inc. (Waltham, MA, USA). Bovine albumin was purchased from Interlab Confiança (Sao Paulo, Brazil). Roswell Park Memorial Institute medium 1640 (RPMI), Dulbecco’s modified Eagle’s medium (DMEM), fetal bovine serum (FBS), penicillin and streptomycin were bought from Gibco-Invitrogen (Thermo Fisher Scientific, USA). Cholesterol, uranyl acetate, dextrose, fluorescein isothiocyanate (FITC), bisbenzimide HOESCHT 33,342 and other reagents were acquired in analytical grade from Sigma-Aldrich Co. (St. Louis, MO, USA).

Mycobacteriophage D29 was kindly provided by Dr. Graham F. Hatfull from the Department of Biological Sciences of the University of Pittsburgh, USA. These phages were previously generated according to the free protocol of PhagesDB available online (http://phagesdb.org/data, accessed on 2 January 2023) [24], and expressed according to Da Silva et al. [25]. The mycobacteriophages D29 (PhageD29) were stored under refrigerated conditions at 4 °C in a phage buffer solution with 10 mM MgSO_4_, 68.5 mM NaCl and 10 mM Tris-HCl at pH 7.6.

### 2.2. Preparation of Nanoliposomes

Nanoliposomes were prepared by the standard technique of lipid film hydration according to Mady et al. [26]. Briefly, a suspension (200 mg) was prepared of cholesterol and phosphatidylcholine (2:3 *w*/*w*) in ether. The lipid film was hydrated with 1:1 *v/v* phage buffer with or without PhageD29 to generate the nanoliposome-encapsulated phage (LipD29) and empty nanoliposome controls (LIP). Then, the nanoliposomes were placed in a rotating incubator at 200 rpm at 37 °C for 100 min.

### 2.3. Characterization of Nanoliposomes

The polydispersity index (PDI) and zeta potential of the samples were diluted in phosphate-buffered saline (PBS) at pH 7, measured using the dynamic light scattering (DLS) technique with a Malvern Zetasizer, where the average size of the nanoliposomes was measured [27]. PDI and zeta potential were obtained using a DTS1060 gold-coated cuvette. Also, for stability, all samples were stored between 2 and 8 °C and read at 24 h and 7, 32, 39, 46, 53 and 60 days after synthesis.

### 2.4. Nanostructure of Nanoliposomes

Nanoliposome morphological analyses were performed using a transmission electron microscope (TEM, JEOL JEM 100 CX II) with 100 kV acceleration and 100,000-fold magnification [28]. First, samples with and without phages were diluted in PBS, pH 7.4 to a final concentration of 0.5 mg/mL and applied to copper grids. Uranyl acetate solution (2%) was added for negative contrast, and after drying at room temperature, photomicrographs were taken.

### 2.5. Mycobacterial Strains

The antimicrobial activity was assessed using the *Mycobacterium tuberculosis* H37Rv ATCC 27,294 (MTB), and latency was determined using *M. tuberculosis* pFCA-luxAB. For the time–kill studies, the strain of *M. smegmatis* mc^2^ 155 was used. The strains were obtained from the collection bank of strains of the TB Research Laboratory from the Faculty of Pharmaceutical Sciences at São Paulo State University (FCF/UNESP). The strains were thawed and activated in 7H9 broth supplemented with bovine albumin fraction V, dextrose, catalase, oleic acid and sodium chloride for 3 weeks (*M. tuberculosis*) or 24 h (*M. smegmatis*) in a shaker with agitation at 200 rpm and 37 °C.

### 2.6. Antimycobacterial Activity

The minimum inhibitory concentration (MIC) assay was performed in 96-well microplates using the microdilution method, following the instructions of the M100 manual, by the Resazurin Microtiter Assay Plate (REMA) method [29,30]. MTB strains were adjusted to 1 McFarland scale (~3 × 10^8^ CFU/mL) and incubated in 7H9 broth supplemented with RIF and isoniazid (INH) at different concentrations (1–0.0003 µg/mL) for 7 days at 37 °C and 5% CO_2_. Mycobacteriophage MIC values were expressed in terms of multiplicity of infection (MOI) to determine the number of phage particles needed to kill bacteria [31], and using tested MOI values from 0.3 to 0.0005. The oxidation and reduction of resazurin were measured using a Synergy H1 microplate reader (BioTek^®^, Winooski, VT, USA) with excitation and emission filters at wavelengths of 530 and 590 nm, respectively. The assays were performed in three independent experiments.

The inhibition of latent strains was determined using the Low-Oxygen Recovery Assay (LORA) method [32]. Briefly, bacteria were cultured in Dubos medium supplemented with albumin, in a closed 2/3 volume flask with little aeration at 37 °C. They were incubated for approximately 15 days until anaerobiosis. After this period, the bacterium is known to have entered a persistent non-replicating state and can be used for the experiment.

### 2.7. Cytotoxic Activity

Cytotoxic activity was evaluated in J774A.1 macrophage (ATCC^®^ Tib-67™, 8 passages), and MRC-5 lung fibroblast (ATCC^®^ CCL-171™, 4 passages) was assessed by the cellular viability test for resazurin (AlamarBlue^®^) [33]. Previously, cells were cultured in DMEM and RPMI media supplemented with 10% (*v*/*v*) fetal bovine serum containing gentamicin (75 µg/mL) and amphotericin (3 μg/mL) under standard conditions (SC, 5% CO_2_ at 37 °C). To determine the half maximal inhibitory concentration (IC_50_), cells (2.5 × 10^5^ cells/mL in RPMI medium and 10^6^ cells/mL in DMEM) were seeded into 96-well microplates and incubated for 24 h under SC. Cells were then treated with 3.2% samples in (1:9 *v*/*v*) phage buffer and PBS, pH 7.4 under SC for 24 h and quantified by the REMA method.

### 2.8. Fluorescence Microscopy

To verify the targeting of the phage formulation in eukaryotic cells, photographs were taken and analyzed with the fluorescence microscope Invitrogen™ EVOS™ FL Auto Imaging System (Thermo Fisher Scientific, USA). Previously, the MRC-5 cells under SC were seeded in 24-well plates with 2.5 × 10^5^ cells/mL and incubated for 24 h under SC. Subsequently, the cells were treated with the (1:10) samples in DMEM for 24 h. Cells were washed with 1X PBS and removed from the surface using trypsin for 2 min. Finally, they were resuspended in DMEM with 10% FBS and the entire content was centrifuged at 2500× *g* for 5 min. Cells were fixed with 200 µL of FITC (10 µM, specific for cell membrane) and 200 µL of bisbenzimide (10 µM, blue fluorescent permeable cell membrane specific for nucleic acid) for 30 min. 4′,6-diamidino-2-phenylindole (DAPI) and FITC were used as controls.

### 2.9. Time–Kill Studies

The time–kill studies for mycobacteriophages encapsulated in nanoliposomes were performed according to NCCLS guidelines [34]. *M. smegmatis* strains were adjusted to 1 McFarland scale (~3 × 10^8^ CFU/mL) and incubated with samples in 7H9 supplemented broth for 48 h at 37 °C and 5% CO_2_. The turbidity related to bacterial growth was measured at 600 nm in triplicate at 8 different times (1, 2, 3, 4, 8, 12, 24 and 48 h).

### 2.10. Intramacrophage Assay

The intramacrophage assay was performed following the method proposed by Snewin et al. [35]. Macrophages were infected with MTB, and after internalization extracellular mycobacteria were removed with amikacin as well as antibiotic residues by washing with PBS. Infected macrophages were treated with the nanoliposomes at 1×, 4× and 10× MIC values. Macrophages were lysed after 72 h of infection with triton X-100 (0.01%) and plated on 7H11 agar plates supplemented for 25 days under SC. The results were expressed in CFU/mL and compared with the growth obtained from untreated wells.

### 2.11. Statistical Analysis

All data were analyzed using an ANOVA test followed by a Newman–Keuls test or Dunnett’s post-test (GraphPad Prism 6.0 software) with significance thresholds of *p* < 0.05. All assays were performed in triplicate.

## 3. Results and Discussion

### 3.1. Physicochemical Results

The formulation was prepared with cholesterol because it reduces the permeability and leakage of the encapsulated volume, and with phosphatidylcholine because it is biocompatible with eukaryotic cells and highly biodegradable [27]. In the preparation of the liposome, phage buffer was used because it has a pH close to 6.0, and values close to 6 have been recommended since they increase the sample stability [36], which is also ideal to keep the phage viable. The DLS readings for size, PDI and zeta potential indicate the stability of the colloidal suspension. A zeta potential close to ±30 mV is expected, since near this value the repulsion between the particles caused by the surface charge occurs, which prevents the aggregation between the vesicles. To indicate the charge, the zeta potential reading demonstrates the stability index of the nanoparticles [37].

During the periodic evaluation of the nanoliposome stability, we found that the nanosystems with or without the mycobacteriophage were in adequate conditions and close to ±30 mV. This makes it clear that the system could be an excellent vehicle to fulfill its action and maintain its stability for several months during storage (Figure 1). TEM photomicrographs confirmed the structure and average size of the nanoliposomes. These results are similar to those previously reported using phages specific for other bacteria [38,39].

According to Nieth et al. [23], to encapsulate mycobacteriophage D29, nanoliposomes must be larger than 500 nm. The TEM images revealed the formation of giant unilamellar vesicles (GUV) for the LIP control, reaching a size greater than 1000 nm [40], and large unilamellar vesicles (LUV) for LipD29 showing a size of 970 nm. Figure 1C shows the phospholipid bilayer surrounding the vesicle and the internal unilamellar aqueous compartment, the total length of the liposome being 1024 nm. Conversely, Figure 1D shows a lipid vesicle with encapsulated material and higher contrast. Furthermore, the inside of the particles was observed, and they were structures of various sizes, including 65 nm which corresponds to the isometric head of mycobacteriophage D29, suggesting phage encapsulation [41]. For intracellular bacteria such as MTB, using larger nanoliposomes favors therapy, since they are captured more quickly by the cells of the mononuclear phagocytic system (macrophages) [42].

The encapsulation efficiency was relatively low (7.1%). The method used for the production of nanoliposomes by thin-film hydration is at a disadvantage compared with other methods due to the low encapsulation efficiency it provides [43]. However, the values obtained are sufficient for phage therapy, since a single mycobacteriophage D29 manages to release an average of 120 phages after cell lysis of macrophages infected with MTB, as reported by David et al. [44].

### 3.2. Cytotoxic Activity

The IC50 values of MRC-5 cells presented in Figure 2 showed no toxicity even at high concentrations of the samples. These results may be attributed to the use of biocompatible compounds such as phosphatidylcholine and cholesterol [45]. In addition, the IC_50_ values in macrophages showed a high rate of inhibition or significant cell death at volumetric concentrations above 60% for the non-encapsulated compounds. However, no significant differences were found in the study of nanoliposomes, suggesting considerable biocompatibility [46].

### 3.3. Fluorescence Microscopy

An in vitro assay was performed to verify the targeting of free and encapsulated phages in treated nanoliposomes on MRC-5 cells. It is known that the release of the material encapsulated in nanoliposomes depends on the fusion (pinocytosis, endocytosis) or exchange of lipids with the cell wall [47]. Figure 3 shows the possible orientation of the phages to the cytoplasm of the cells by fluorescence microscopy, as observed in the cellular membrane structures with adhered circular characteristics, suggesting the internationalization of the formulations in the cytoplasm.

### 3.4. Antimycobacterial Results

According to Jorquera et al. [48], a higher MOI generates greater bacterial lysis. In this sense, the inhibitory activity of the free and encapsulated D29 phages showed a slight response against active MTB (Table 1). These results demonstrate that phage D29 encapsulation within nanoliposomes does not produce changes in the bacterial infection during treatment. However, a comparison with a prolonged treatment such as that with RIF or INH would further highlight the benefits of D29 because the decrease in the bacterial load is remarkable with only one dose of treatment, as shown in the growth plates (Figure 4). This is really advantageous because cytotoxicity or adverse effects caused by prolonged use of these drugs would be avoided.

On the other hand, the treatment using the LORA method conducted to visualize its effectiveness against latent tuberculosis shows even better results. It reveals that more viruses are capable of infecting MTB. A pre-treatment could be considered for people with a compromised immune system since low doses could eliminate high bacterial loads. In addition, greater efficacy is obtained at low doses of phages, as this is crucial for phages to hide from the immune system and avoid being removed before reaching the infection site [49]. The unloaded liposomes did not show antimicrobial activity, evidencing normal MTB growth.

Nanoliposomes would also allow the action of these bacteriophages to be even more effective because the fluorescence microscopy studies corroborated the insertion of the nanoliposomes and, consequently, of the phages in the MRC-5 cells infected with MTB. This intracellular therapy may be considered an advance in the treatment of multi- and extremely drug-resistant strains [50].

Studies based on lytic phages have demonstrated the effectiveness of these viruses on widely resistant bacterial strains of *Acinetobacter baumannii* and their ability to modify the bacterial wall, one of the results being the loss of resistance [51]. The study used phage-resistant *A. baumannii* capsular polysaccharides as receptors and they were treated with ΦFG02 and ΦCO01, eliminating the bacterial load in blood when applied in vitro and in vivo using mutant-infected mice. This effect was possible due to the loss of the bacterial capsule, which is responsible for virulence in *A. baumanni.* Phages attach to specific receptors that may be on the surface of the bacterial membrane, or even on pili and flagella [52]. The results revealed the sensitization of the strains against beta-lactams, the complement system and alternative phages [51].

### 3.5. Time–Kill Results

Studies carried out in various periods made it possible to identify several obstacles during phage therapy, since these phages at high concentrations are quickly eliminated by the immune system [53]. The results showed the high activity of the encapsulated phages against *M. smegmatis* with a MOI of 0.02 after 48 h of exposure. The intention was to show that the optimal concentration for the inhibition of planktonic *M. smegmatis* was 0.2 MOI, since the higher the MOI, the better the activity against MTB (Figure 5A). It was also shown that the phages or the materials used for encapsulation or maintenance did not influence the antimicrobial activity (Figure 5B). The D29 phage against *M. smegmatis* managed to prevent the growth of the mycobacteria, generating a significant reduction of 5 Log_10_ CFU/mL for a MOI of 0.2, and of 3 Log_10_ CFU/mL for a MOI of 0.02. The data show that the multiplicities of infection used were efficient to demonstrate the lytic action of mycobacteriophage D29 against the *M. smegmatis* strain.

The results presented in Figure 5C show that MIC excess values from free phage D29 greater than 10× are not recommended for an efficient activity. When 4× MIC values were used, the intramacrophage antimycobacterial activity of 35% was evident. However, this was not the case with 10× MIC (MOI = 0.01), where no inhibition of mycobacterial growth occurred. Some authors describe this phenomenon as an immunological response of macrophages against phages, so they recommend using lower MOI values during phage treatment [49,54]. Likewise, Xiong et al. [55] showed phage D29 activity against MTB in the macrophage infection phase (BALB/C peritoneal) in less time and with greater efficiency than control drugs. The results demonstrate the efficacy of the liposome as a phage detection vehicle within MTB-infected macrophages and 53.5% inhibition at MOI = 0.01, that is, 10× MIC values for the formulation of nanoliposomes with encapsulated phages compared with free phages.

The two drugs used as reference in these experiments were rifampicin and isoniazid where the main reason for the search for alternative treatments focuses on their resistance. Resistance to RIF in MTB is generated mainly by genetic mutations in the rpoB gene, which encodes the β subunit of the RNA polymerase (RNAP) of the bacteria, due to the absence of plasmids and low frequency of recombination in this bacteria [56]. RIF inhibits RNA synthesis by binding to RNAP, but mutations in rpoB can alter the structure of the β subunit, preventing RIF binding and causing resistance. About 90% to 100% of RIF-resistant MTB isolates have mutations in rpoB, with point mutations being the most common [57]. Predominant mutations in a specific region of the rpoB gene vary in frequency among different MTB strains, suggesting a correlation between strain genetics and RIF resistance. Machine learning tools have been developed to predict RIF resistance based on these mutations [58]. On the other hand, INH is a prodrug that requires activation by the KatG catalase–peroxidase enzyme. Once activated, the resulting metabolites interact with nicotinamide adenine dinucleotide (NAD+) forming an INH-NAD adduct, which subsequently binds to the NADH-dependent enoyl-ACP reductase enzyme InhA. This binding inhibits the production of mycolic acid, a crucial cell wall component in MTB, resulting in inhibition of cell wall synthesis and causing cell death. INH resistance (INH-R) in several strains is attributed to mutations in genes essential for cell wall synthesis [59]. The mutations are mainly found in the katG gene, the inhA gene and its promoter or in the oxiR-ahpC region. It has been observed that strains with deletions in the katG gene show greater resistance to INH compared with those with mutations in inhA or its promoter. Furthermore, it has been identified that overexpression of INH inactivators or efflux pumps contributes to resistance to INH, expanding the understanding of the mechanisms of resistance to this prodrug in MTB [60].

These two drugs have receptors and enzymatic ligands, which generate specificity as well as bacteriophages. However, mycobacteriophages possess sophisticated machinery designed for the efficient lysis of mycobacteria. A central component of this machinery is the genetic encoding of lysines or endolysins (LysA), which are specialized enzymes with the ability to break down the peptidoglycan (PG) layer, a crucial element of the mycobacterial cell envelope. The catalytic activity of these endolysins is specifically oriented towards the β-1,4-glycosidic bonds between *N*-acetylmuramic acids and *N*-acetylglucosamine present in the peptidoglycan matrix, which facilitates the destabilization and rupture of this structure vital for the integrity of the host cell [61]. However, the lysis machinery in mycobacteriophages is not restricted to the action of endolysins; it also incorporates holins, which regulate lysis time by facilitating the release of endolysins to the periplasmic space, and possibly spanins, which contribute to the disruption of the outer membrane [62]. In fact, recent research on the physiology of mycobacteriophage D29 reveals that, although holin is not essential for phage viability, it is crucial for timely and efficient lysis of the host cell, as well as for the propagation of phage progeny [62].

Interestingly, the results showed effective activity against latent MTB, which is beneficial given the specificity of this system. Latent MTB, unlike active MTB, has several survival properties in extreme conditions, allowing it to enter a state of dormancy capable of surviving in the absence of nutrients and oxygen, latent MTB being the most difficult phase of tuberculosis to detect and treat [63]. Recently our laboratory reported that nanoparticulate systems would be capable of eradicating multi- and extremely RIF-resistant MTB through multiple attacks, which would be favorable for future research into nanosystems and the use of conjugation or surface modification to further improve the targeted administration [64]. Previous studies reported the efficiency of liposomes in MTB strains, corroborating the effectiveness of the use of phages for the treatment of active and latent MTB [65,66].

## 4. Conclusions

Bacteriophages are excellent bacteria hunters due to their specificity and selection. We demonstrated that the D29 bacteriophage had effectiveness and affinity for *Mycobacerium tuberculosis* and *smegmatis*. It also showed its effectiveness in latent strains through the LORA assay. The encapsulation of D29 allows not only the transport of phages but also their cellular insertion as well as their prevalence for a controlled treatment, avoiding immunodestructive attacks caused when excess phages are administered. Phage-loaded nanoliposomes would be an effective treatment tool for one of the world’s deadliest bacteria, *Mycobacterium tuberculosis*.

## Figures and Tables

**Figure 1 diseases-11-00150-f001:**
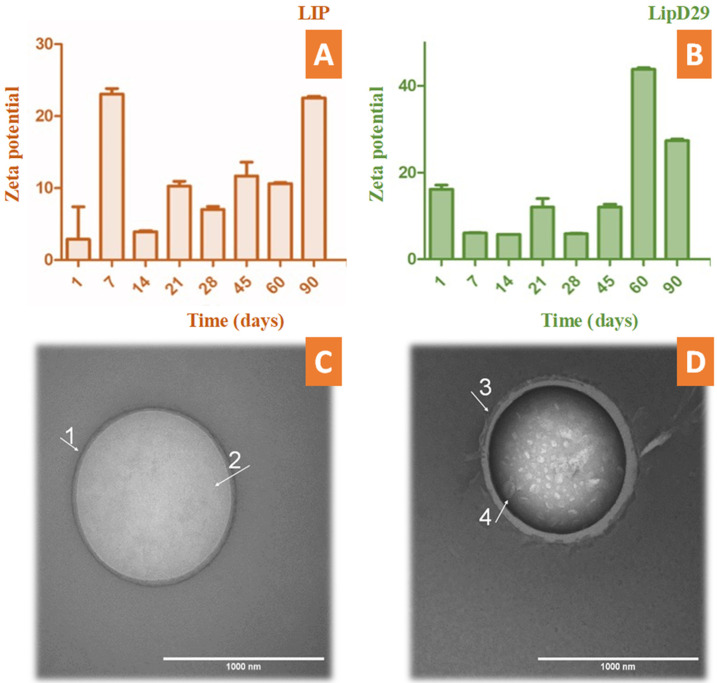
Stability study by zeta potential over time up to 90 days of (**A**) free and (**B**) loaded nanoliposomes. TEM photomicrographs obtained with uranyl staining (2%) at 100,000× magnification. (**C**) LIP (GUV) of 1024 nm in diameter delimited by a (1) phospholipid bilayer and containing (2) an aqueous inner compartment; (**D**) LipD29 (LUV) with a diameter of 970 nm also bounded by (3) a phospholipid bilayer and containing an interior compartment with encapsulated structures approximately 65 nm in width (4).

**Figure 2 diseases-11-00150-f002:**
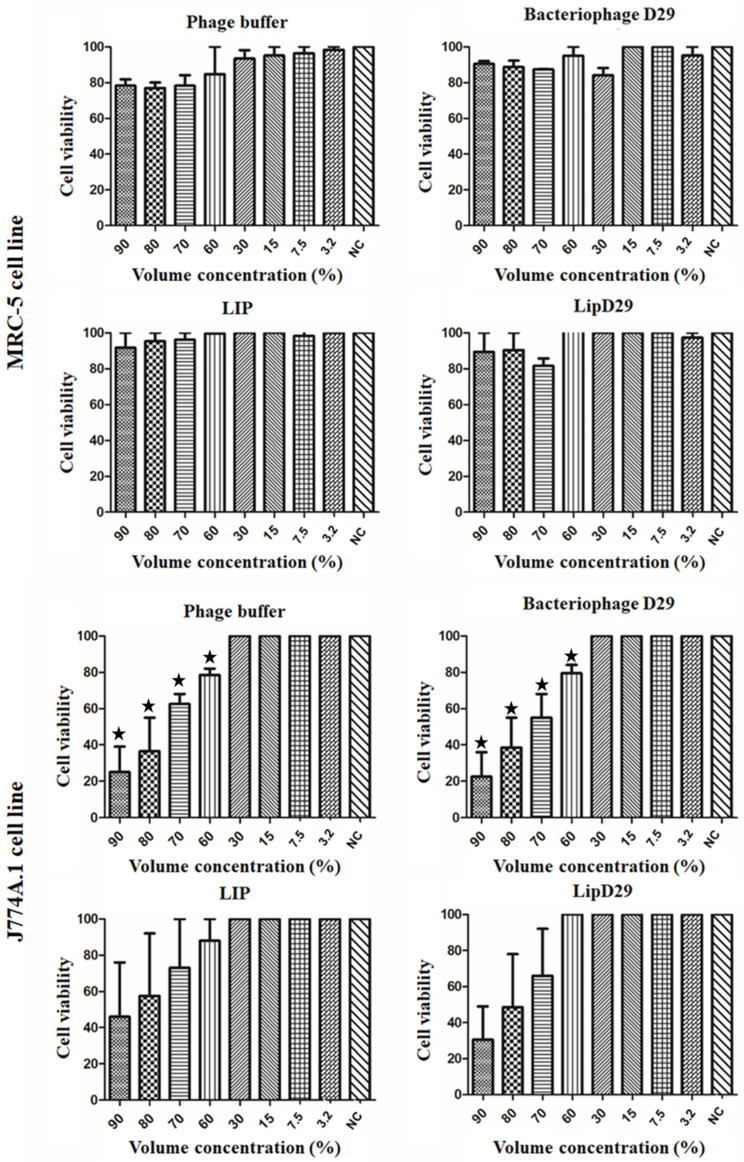
Cell viability (IC_50_) for lung macrophages (J774A.1) and fibroblast (MRC-5) cells. Values presented as the mean of two independent tests (mean ± standard error) and statistical analysis of variance by the One-Way test followed by Dunnett’s, with a significance level of 5% (*p* < 0.05). * There is a significant difference.

**Figure 3 diseases-11-00150-f003:**
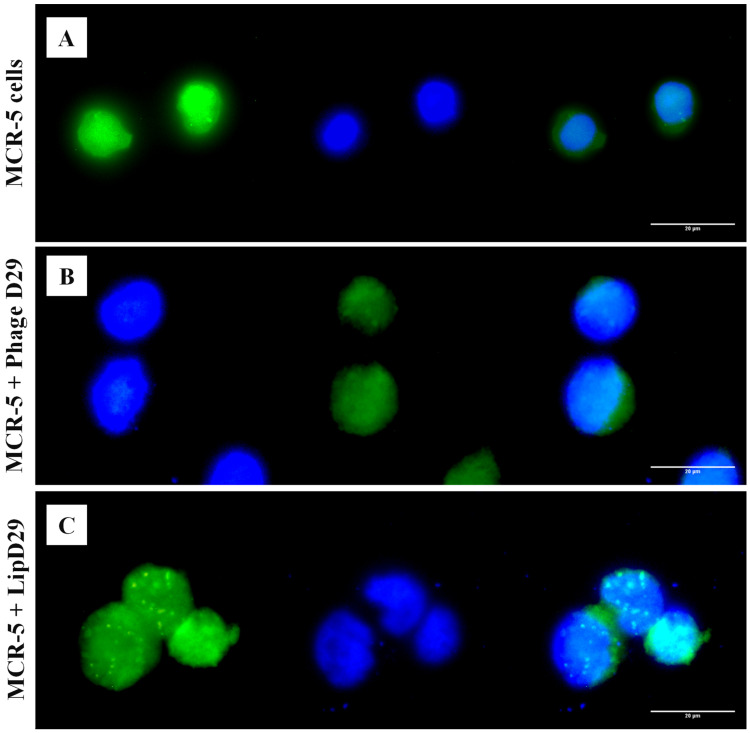
Fluorescence microscopy with free and encapsulated mycobacteriophage D29 using MRC-5 cell lines. (**A**) MRC-5 cell control. (**B**) Cells with free D29 phage. (**C**) MRC-5 cells incubated with D29 phage encapsulated in nanoliposomes.

**Figure 4 diseases-11-00150-f004:**
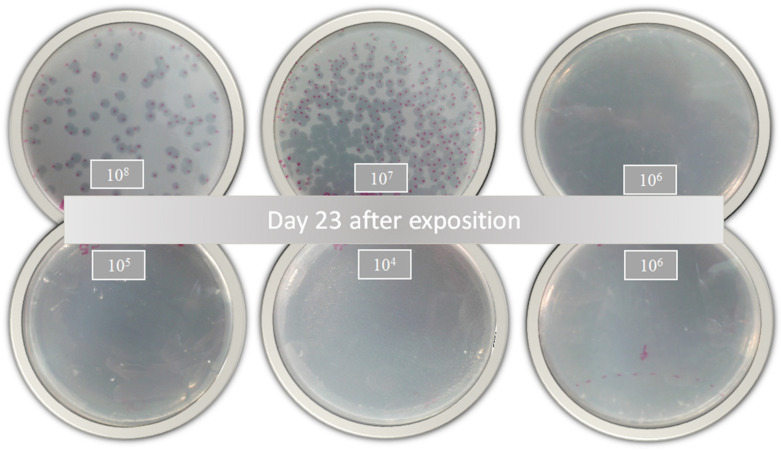
Bacterial growth of *Mycobacterium tuberculosis* 23 days after treatment with nanoliposomes loaded with phage D29 exposed for 24 h.

**Figure 5 diseases-11-00150-f005:**
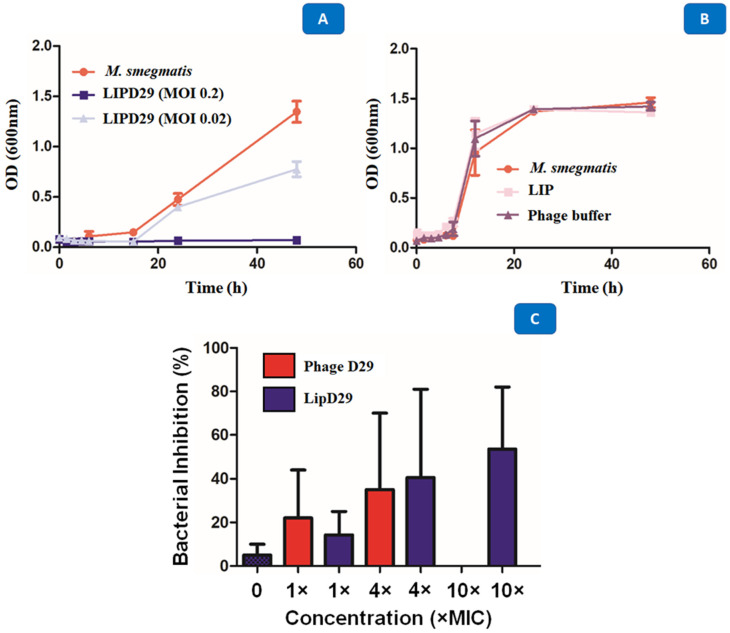
(**A**) Growth curve with *M. smegmatis* and mycobacteriophage D29 at a multiplicity of infection of 0.2 and 0.02. (**B**) Time–kill with *M. smegmatis*, phage buffer and empty nanoliposome. (**C**) Intramacrophage assay with mycobacteriophage D29 free and encapsulated in nanoliposomes at concentrations of 1×, 4×, 10× MIC_90_ values against MTB.

**Table 1 diseases-11-00150-t001:** Minimum inhibitory concentration (MIC_90_) assay against *Mycobacterium tuberculosis* (H_37_Rv) in an active metabolic state (REMA) and dormant stage (LORA).

Sample	REMA		LORA	
	MIC_90_ (µg/mL)	MOI	MIC_90_ (µg/mL)	MOI
Rifampicin	0.06 ± 0.05	-	1.2 ± 0.81	-
Isoniazid	0.03 ± 0.02	-	152.4 ± 3.14	-
PhageD29	-	0.001	-	>2.7
LipD29	-	0.001 ± 0.0008	-	>2.7

## Data Availability

The data presented in this study are available on request from the corresponding author. The data are not publicly available due to national regulations.

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
