# Peer review of "Activity of Bacteriophage D29 Loaded on Nanoliposomes against Macrophages Infected with Mycobacterium tuberculosis"

_diseases, 2023, doi:10.3390/diseases11040150_

Round 1

Reviewer 1 Report

Dear  Editor and authors,

The manuscript entitled "Activity of Bacteriophage D29 Loaded on Nanoliposomes against Macrophages Infected with Mycobacterium tuberculosis" by Silva, A.P. B. et al., submitted to Diseases (MDPI), describes the preparation of mycobacteriophage D29 nanoliposomes and their application against dormant MTB in macrophages. Nanoliposomes (with and without bacteriophage D29) were prepared from cholesterol and phosphatidylcholine (in the presence and absence of a phage buffer solution) to generate the nanoliposome-encapsulated phage (LipD29) and empty nanoliposome controls (LIP). The polydispersity index (PDI) and zeta potential were measured using dynamic light scattering (DLS) technique. The DLS readings for size, PDI, and zeta potential indicate the stability of the colloidal suspension. During the periodic evaluation of nanoliposome stability, the authors found that the zeta potential of nanosystems with or without mycobacteriophage is close to ±30 mV, providing a good vehicle to fulfill its action and maintain its stability for several months during storage.

1st comment: The authors do not mention the conditions used for measuring zeta potential (salt content, pH). Please include this information.

Morphological analysis of nanoliposomes was performed using TEM. The TEM images revealed the formation of giant unilamellar vesicles (GUV) for the LIP control, reaching a size greater than 1000 nm, and large similar unilamellar vesicles (LUV) for LipD29, allowing encapsulation of D29. The cytotoxic activity was evaluated with J774A.1 macrophage and MRC-5 lung fibroblast. The IC50 values of MRC-5 and J774A.1 cells showed no toxicity at high concentrations of the samples. Fluorescence microscopy was performed to verify the targeting of the phage formulation in eukaryotic cells and showed the internalization of the phage formulation.

The antimicrobial activity was confirmed using Mycobacterium tuberculosis H37Rv ATCC 27294 by the Resazurin Microtiter Assay Plate (REMA) method. Mycobacteriophage MIC values were expressed in terms of the multiplicity of infection (MOI) to determine the number of phage particles needed to kill bacteria (Vipra et al., 2013), using tested MOI-values from 0.3 to 0.0005. A higher MOI generates greater bacterial lysis. The inhibitory activity of the free and encapsulated D29 phages showed a slight response against active MTB (Table 1) and demonstrated that phage D29 encapsulation within nanoliposomes does not produce changes in the bacterial infection during treatment. The authors claim that a comparison with prolonged treatment such as that with rifampicin or isoniazid would further highlight the benefits of D29 because the decrease in the bacterial load is remarkable with only one dose of treatment, avoiding prolonged use of these drugs. The inhibition of latent strains was determined using the Low-Oxygen Recovery Assay (LORA) to visualize its effectiveness against latent tuberculosis, revealing that more viruses are capable of infecting MTB. Greater efficacy is obtained at low doses of phages, as this is crucial for phages to evade the immune system and avoid being removed before reaching the infection site.

2nd comment: Please correct any typographical errors in Table 1.

Time-kill studies for mycobacteriophage encapsulated in nanoliposomes were performed according to NCCLS guidelines (Wikler, 2006) on M. smegmatis. It was possible to identify several obstacles during phage therapy, as these phages at high concentrations are quickly eliminated by the immune system. The results demonstrate the efficacy of the liposome as a phage detection vehicle within MTB-infected macrophages for the formulation of nanoliposomes with encapsulated phages compared to free phages.

3rd comment: I am not sure if the authors cited all the relevant literature. Check! I found a recent article dealing with a very similar topic about the application of very similar in content liposomal form of D29 against TB (Lepenkova, M. B et al. Bulletin of Experimental Biology and Medicine, Vol. 169, No. 3, July 2020 IMMUNOLOGY AND MICROBIOLOGY) that is not cited. Comparing to  Lepennkova et al. what is innovation of this work? Authors should discuss and compare.

The authors demonstrated that the D29 bacteriophage had effectiveness and affinity for M. tuberculosis and M. smegmatis, showing their effectiveness in latent strains through the LORA assay, as well as the encapsulation of D29 in nanoparticles. These nanoparticles allow not only the transport of phages but also avoid immunodestructive attacks caused by excess phages when administered. Phage treatment may be an interesting innovative solution for MTB treatment.  I found  work very interesting but not completely innovative and thus recommend a major corrections in this respect.

Thank you for your collaboration.

Author Response

REVIEWER 1

The manuscript entitled "Activity of Bacteriophage D29 Loaded on Nanoliposomes against Macrophages Infected with Mycobacterium tuberculosis" by Silva, A.P. B. et al., submitted to Diseases (MDPI), describes the preparation of mycobacteriophage D29 nanoliposomes and their application against dormant MTB in macrophages. Nanoliposomes (with and without bacteriophage D29) were prepared from cholesterol and phosphatidylcholine (in the presence and absence of a phage buffer solution) to generate the nanoliposome-encapsulated phage (LipD29) and empty nanoliposome controls (LIP). The polydispersity index (PDI) and zeta potential were measured using dynamic light scattering (DLS) technique. The DLS readings for size, PDI, and zeta potential indicate the stability of the colloidal suspension. During the periodic evaluation of nanoliposome stability, the authors found that the zeta potential of nanosystems with or without mycobacteriophage is close to ±30 mV, providing a good vehicle to fulfill its action and maintain its stability for several months during storage.

1st comment: The authors do not mention the conditions used for measuring zeta potential (salt content, pH). Please include this information.

R: The authors appreciate your comment. We have added the conditions in the materials and methods section.

Morphological analysis of nanoliposomes was performed using TEM. The TEM images revealed the formation of giant unilamellar vesicles (GUV) for the LIP control, reaching a size greater than 1000 nm, and large similar unilamellar vesicles (LUV) for LipD29, allowing encapsulation of D29. The cytotoxic activity was evaluated with J774A.1 macrophage and MRC-5 lung fibroblast. The IC50 values of MRC-5 and J774A.1 cells showed no toxicity at high concentrations of the samples. Fluorescence microscopy was performed to verify the targeting of the phage formulation in eukaryotic cells and showed the internalization of the phage formulation.

The antimicrobial activity was confirmed using Mycobacterium tuberculosis H37Rv ATCC 27294 by the Resazurin Microtiter Assay Plate (REMA) method. Mycobacteriophage MIC values were expressed in terms of the multiplicity of infection (MOI) to determine the number of phage particles needed to kill bacteria (Vipra et al., 2013), using tested MOI-values from 0.3 to 0.0005. A higher MOI generates greater bacterial lysis. The inhibitory activity of the free and encapsulated D29 phages showed a slight response against active MTB (Table 1) and demonstrated that phage D29 encapsulation within nanoliposomes does not produce changes in the bacterial infection during treatment. The authors claim that a comparison with prolonged treatment such as that with rifampicin or isoniazid would further highlight the benefits of D29 because the decrease in the bacterial load is remarkable with only one dose of treatment, avoiding prolonged use of these drugs. The inhibition of latent strains was determined using the Low-Oxygen Recovery Assay (LORA) to visualize its effectiveness against latent tuberculosis, revealing that more viruses are capable of infecting MTB. Greater efficacy is obtained at low doses of phages, as this is crucial for phages to evade the immune system and avoid being removed before reaching the infection site.

2nd comment: Please correct any typographical errors in Table 1.

R: The authors appreciate your comment. We have corrected the writing error.

Time-kill studies for mycobacteriophage encapsulated in nanoliposomes were performed according to NCCLS guidelines (Wikler, 2006) on M. smegmatis. It was possible to identify several obstacles during phage therapy, as these phages at high concentrations are quickly eliminated by the immune system. The results demonstrate the efficacy of the liposome as a phage detection vehicle within MTB-infected macrophages for the formulation of nanoliposomes with encapsulated phages compared to free phages.

3rd comment: I am not sure if the authors cited all the relevant literature. Check! I found a recent article dealing with a very similar topic about the application of very similar in content liposomal form of D29 against TB (Lepenkova, M. B et al. Bulletin of Experimental Biology and Medicine, Vol. 169, No. 3, July 2020 IMMUNOLOGY AND MICROBIOLOGY) that is not cited. Comparing to  Lepennkova et al. what is innovation of this work? Authors should discuss and compare.

R: The authors thank you for your valuable comment. We have tried to locate the referred manuscript but unfortunately we have not obtained it. However, this manuscript stands out for the appropriate use of bacteriophage for a possible application in Mycobacterium tuberculosis in the latent phase; Since in this phase, this bacteria is capable of modifying its own structure to evade the immune system for years, until the patient suffers a loss of the immune system. The results highlight its application at a single dose, a superior result than drugs already used in the latent phase such as Isoniazid or Rifampicin.

The authors demonstrated that the D29 bacteriophage had effectiveness and affinity for M. tuberculosis and M. smegmatis, showing their effectiveness in latent strains through the LORA assay, as well as the encapsulation of D29 in nanoparticles. These nanoparticles allow not only the transport of phages but also avoid immunodestructive attacks caused by excess phages when administered. Phage treatment may be an interesting innovative solution for MTB treatment.  I found  work very interesting but not completely innovative and thus recommend a major corrections in this respect.

We appreciate your comment, and we highlight that we are preparing other types of more sophisticated nanosystems for this purpose; However, in this research or current form we cannot include additional results due to lack of funding to work on a BSL 3 with infected cells, our project was concluded and closed.

Reviewer 2 Report

The overall study presents a new therapeutic strategy to discover new antimycobacterial agents specially targeted to M. tb and M. smegmatis using bacteriophage science and technology. Before the manuscript can be accepted for publication, the authors must address the following comments.

1. In the abstract, the authors did not substantially highlight the methods used.

2. Also, some results are missing congruent to the methods used in the study. The authors are asked to supplement and highlight results and provide thereof important generalizations.

3. In the intro, the authors are also asked to provide key methods in their objectives.

4. In the time kill assay, why only use M. smegmatis? How about M. tb? 

Minor editing is required. Also formats ie for species scientific nomenclature

Author Response

REVIEWER 2

The overall study presents a new therapeutic strategy to discover new antimycobacterial agents specially targeted to M. tb and M. smegmatis using bacteriophage science and technology. Before the manuscript can be accepted for publication, the authors must address the following comments.

  1. In the abstract, the authors did not substantially highlight the methods used.

The authors appreciate your comment and suggestion. We have included the methods as requested.

  1. Also, some results are missing congruent to the methods used in the study. The authors are asked to supplement and highlight results and provide thereof important generalizations.

The authors appreciate your suggestion. Currently it is impossible for us to increase more experiments since it is a concluded project and we do not have sufficient resources to resume the project. Additionally, we know that these results can help corroborate the efficient activity of bacteriophages, mainly in Mycobacterium tuberculosis in its latent phase. As we recently reported (Carnero et al., 2023), there are no treatments yet approved by the FDA to counterattack latent MTB, there is a long path to these events. The main preventive method for people with HIV is isoniazid, previously using a rapid tuberculin test (not entirely effective). We hope that with this manuscript we can show that bacteriophages can be an excellent tool, both for single-dose treatment and for diagnosis due to their precision. We have added some highlights at the end of the manuscript.

https://doi.org/10.3390/pharmaceutics15102409

  1. In the intro, the authors are also asked to provide key methods in their objectives.

The authors appreciate your suggestion. State-of-the-art was included in the manuscript.

  1. In the time kill assay, why only use M. smegmatis? How about M. tb?

The authors thank you for your comment. The study has been carried out on M. smegmatis due to the growth speed of the bacteria. Mycobacterium tuberculosis (MTB) compared to smegmatis, grows (up to McFarlad values of 1) for approximately 3 weeks, if it has not been contaminated during transit. Additionally, TimeKill MTB studies must be done for 21 days. And unfortunately in Brazil during the development of the experiments there were several cuts in grants to sustain the project (during the COVID pandemic) and Mycobacterium tuberculosis involves the use of a BSL3 highly equipped with negative pressure and expensive PPE. In the midst of the pandemic, the laboratory paralyzed its activities. However, all our works have currently been evaluated mainly in MTB additional to M. smegmatis and M. bovis (Galleria mellonella).

Reviewer 3 Report

The antibacterial activity of the mycobacteriophage D29 during transport and its stability in the treatment against active and latent Mycobacterium tuberculosis have been investigated by Ana P. B. Silva et al. Effective cell connectivity was demonstrated by phage nanoencapsulation.

With values more than 90%, phage nanoencapsulation demonstrated effective cell inter- 25 nalization to induce MTB clearance. As a result, it was demonstrated that nanotechnology can help degradation-sensitive chemicals function in order to accomplish better therapy and avoid the immune reaction to phages while receiving treatment.

Following the completion of the minor adjustments listed below, the work can be accepted for publication.

1- Figure 1 can be omitted because it is not required. It is sufficient to use an acceptable text.

2. In accordance with the directions in the journal, the text references should be replaced by numbers.

3. The figures' numbering has to be changed. Figure 5 precedes Figure 3 in the text. Figure 3 should therefore be used in its stead.

Author Response

REVIEWER 3

The antibacterial activity of the mycobacteriophage D29 during transport and its stability in the treatment against active and latent Mycobacterium tuberculosis have been investigated by Ana P. B. Silva et al. Effective cell connectivity was demonstrated by phage nanoencapsulation.

With values more than 90%, phage nanoencapsulation demonstrated effective cell inter- 25 nalization to induce MTB clearance. As a result, it was demonstrated that nanotechnology can help degradation-sensitive chemicals function in order to accomplish better therapy and avoid the immune reaction to phages while receiving treatment.

Following the completion of the minor adjustments listed below, the work can be accepted for publication.

1. Figure 1 can be omitted because it is not required. It is sufficient to use an acceptable text.

The authors appreciate your comment. Figure 1 has been removed.

2. In accordance with the directions in the journal, the text references should be replaced by numbers.

The authors appreciate your comment. As requested by the reviewer, we have updated the references to the journal standards.

3. The figures' numbering has to be changed. Figure 5 precedes Figure 3 in the text. Figure 3 should therefore be used in its stead.

We regret the error in the numbering of the Figures. It has been corrected in the new version.

Round 2

Reviewer 1 Report

              Dear Editor and Authors,

The authors responded favorably to most of my comments. I noticed they added more references, but unfortunately, the section for references is not included, so I couldn't view them. I am providing the full reference information (Ref No1) and the DOI information about the work I find of great interest for their manuscript. Additionally, I am suggesting one more recent work (Ref No2) by the same author, Vladimirski, M. A. et al. I believe that the work published in these two references bears significant similarities to the authors' work and should be cited and discussed if not already included in the new version. This work does not focus on nanoliposomes but Vladimirski, M. A. et al. also utilize the micobacteriophage D29 for MTB treatment. The chemical composition of the liposomal formulation is practically the same or very similar, and they deal with delivery to tuberculosis granuloma, which is inherently connected to latent TB and bacteria. I am positive that some discussion of this work must be included.

1) Lepenkova, M. B., Alyapkina, Y. S., Vladimirski M. A. “Bactericidal Activity of Liposomal Form of Lytic Mycobacteriophage D29 in Cell Models of Tuberculosis Infection In Vitro” Bull Exp Biol Med. 2020 Jul;169(3):361-364. doi: 10.1007/s10517-020-04887-6.

https://doi.org/10.1007/s10517-020-04887-6

2) Avdeev, V.V., Kuzin, V.V., Vladimirsky, M. A.,Vasilieva, I. A. “Experimental Studies of the Liposomal Form of Lytic Mycobacteriophage D29 for the Treatment ofTuberculosis Infection”

Microorganisms 2023, 11(5), 1214;

https://doi.org/10.3390/microorganisms11051214

Thank you for your collaboration.

Author Response

The authors appreciate the reviewer's comments. We have included the references in the manuscript. We believe that the intention to publish these results may show an advance for the use of nanotechnology in the administration of phages for a better treatment against MTB.

Round 3

Reviewer 1 Report

Dear Editor and Authors,

The authors responded all comments favourably. The manuscript now can be accepted for publishing.

Thank you for your collaboration.